# The C4 Protein of TbLCYnV Promotes SnRK1 β2 Degradation Via the Autophagy Pathway to Enhance Viral Infection in *N. benthamiana*

**DOI:** 10.3390/v16020234

**Published:** 2024-02-01

**Authors:** Xinquan Li, Min Zhao, Wanyi Yang, Xueping Zhou, Yan Xie

**Affiliations:** 1State Key Laboratory of Rice Biology and Breeding, Ministry of Agriculture Key Laboratory of Molecular Biology of Crop Pathogens and Insect Pests, Key Laboratory of Biology of Crop Pathogens and Insects of Zhejiang Province, Institute of Biotechnology, Zhejiang University, Hangzhou 310058, China; 22016079@zju.edu.cn (X.L.); zzhou@zju.edu.cn (X.Z.); 2State Key Laboratory for Biology of Plant Diseases and Insect Pests, Institute of Plant Protection, Chinese Academy of Agricultural Sciences, Beijing 100193, China

**Keywords:** SnRK1, NbSnRK1 β2, TbLCYnV C4, interaction, degradation, autophagy pathway

## Abstract

Geminiviruses are a group of single-stranded DNA viruses that have developed multiple strategies to overcome host defenses and establish viral infections. Sucrose nonfermenting-1-related kinase 1 (SnRK1) is a key regulator of energy balance in plants and plays an important role in plant development and immune defenses. As a heterotrimeric complex, SnRK1 is composed of a catalytic subunit α (SnRK1 α) and two regulatory subunits, β and γ. Previous studies on SnRK1 in plant defenses against microbial pathogens have mainly focused on SnRK1 α. In this study, we validated the interaction between the C4 protein encoded by tobacco leaf curl Yunnan virus (TbLCYnV) and the regulatory subunit β of *Nicotiana benthamiana* SnRK1, i.e., NbSnRK1 β2, and identified that the Asp22 of C4 is critical for TbLCYnV C4–NbSnRK1 β2 interactions. *NbSnRK1 β2* silencing in *N. benthamiana* enhances susceptibility to TbLCYnV infection. Plants infected with viral mutant TbLCYnV (C4^D22A^), which contains the mutant version C4 (D22A) that is incapable of interacting with NbSnRK1 β2, display milder symptoms and lower viral accumulation. Furthermore, we discovered that C4 promotes NbSnRK1 β2 degradation via the autophagy pathway. We herein propose a model by which the geminivirus C4 protein causes NbSnRK1 β2 degradation via the TbLCYnV C4–NbSnRK1 β2 interaction to antagonize host antiviral defenses and facilitates viral infection and symptom development in *N. benthamiana*.

## 1. Introduction

Geminiviruses are a family of single-stranded DNA (ssDNA) viruses that infect many plant species and cause serious diseases in important crops [1,2,3,4]. The family *Geminiviridae* is currently divided into 14 genera based on the genome organization, host range, phylogenetic relationship, and insect vector: *Becurtovirus*, *Begomovirus*, *Capulavirus*, *Citlodavirus*, *Curtovirus*, *Eragrovirus*, *Grablovirus*, *Maldovirus*, *Mastrevirus*, *Mulcrilevirus*, *Opunvirus*, *Topilevirus*, *Topocuvirus*, and *Turncurtovirus* [5,6]. As the largest genus with 445 species, begomoviruses are transmitted by whiteflies and have caused serious threats to the production of many vegetable and fiber crops [5,7]. Begomoviruses are divided into two groups: monopartite begomovirus, which includes only one circular genome, and bipartite begomovirus, which consists of two circular genomes [4,8]. Tobacco leaf curl Yunnan virus (TbLCYnV) is a typical monopartite begomovirus and can produce severe upward leaf curling and vein thickening symptoms in plants [9].

During long-term evolution, viruses have developed elaborate strategies like encoding multifunctional protein C4 to disturb plant defenses. C4 is the smallest protein encoded by begomovirus and has been described as a symptom determinant [10]. Plants expressing tomato leaf curl Yunnan virus (TLCYnV) C4 develop virus-like symptoms that consist of severe leaf curling and distortion [11]. The S-acylated form of C4 in beet severe curly top virus (BSCTV) interacts with CLAVATA 1 (CLV1), an important receptor kinase in meristem maintenance, and interferes with the binding of CLV1 to its target WUSCHEL, resulting in abnormal cell cycles, forming symptoms similar to plant proliferation during viral infection [12]. Some C4/AC4 proteins have been described to function as suppressors of transcriptional gene silencing (TGS) or post-transcriptional gene silencing (PTGS) [10]. The C-terminal region of C4 in tomato leaf curl virus-Australia (ToLCV) seems essential for suppressing RNA silencing [13]. Geminivirus C4 proteins can counter plant defense responses, which are not limited to antiviral gene silencing during viral infection [14]. TLCYnV C4 can impair a hypersensitivity-induced response 1 (HIR1)-mediated hypersensitive response and escape host resistance by interfering with the self-interaction of HIR1 and promoting HIR1 degradation [15].

Sucrose non-fermenting-1-related kinase 1 (SnRK1) lies at the heart of metabolic homeostasis in plants and is crucial for normal development and response to stress [16]. SnRK1 has been reported to operate as a heterotrimeric complex consisting of a catalytic subunit α (SnRK1 α) and two regulatory subunits (β and γ) [17]. SnRK1 not only acts as a molecular fuel meter to sense the cellular energy status but also plays an important role in plant immune defenses [18]. The effector protein AvrBsT from *Xanthomonas campestris pv. vesicatoria* (*Xcv*) interacts with SnRK1 α in plants, suggesting that targeting SnRK1 might be important for *Xcv* pathogenicity [19]. The upregulation or downregulation of SnRK1 α leads to the resistance or susceptibility to the rice blast pathogen *Magnaporthe oryzae* and *Xanthomonas oryzae pv. oryzae* in rice, respectively [20]. SnRK1 α interacts with wheat protein (Ta-FROG), promoting host resistance to *Fusarium graminearum* and mycotoxin deoxynivalenol (DON) [21].

Nowadays, studies on SnRK1 in plant defenses against microbial pathogens have focused on SnRK1 α. In this study, we tested the interaction between TbLCYnV C4 and NbSnRK1 β2 and identified that the Asp22 of TbLCYnV C4 was critical for its interaction with NbSnRK1 β2. We found that *NbSnRK1 β2* silencing in *Nicotiana benthamiana* enhanced susceptibility to TbLCYnV infection. The mutant virus TbLCYnV (C4^D22A^), containing the mutated-C4, which is unable to interact with NbSnRK1 β2, produced milder symptoms and slight viral accumulation than those of the wild TbLCYnV in *N. benthamiana*. In addition, we found that TbLCYnV C4 promoted NbSnRK1 β2 degradation via the autophagy pathway. In summary, we propose that the geminivirus C4 protein caused NbSnRK1 β2 degradation via the TbLCYnV C4–NbSnRK1 β2 interaction through the autophagy pathway to reduce plant defenses and promoted viral infection and symptom development in *N. benthamiana*, which has not been previously reported.

## 2. Materials and Methods

### 2.1. Plant Material and Growth Conditions

Transgenic *N. benthamiana* plants expressing the nuclear marker *H2B-RFP* (full-length red fluorescent protein fused to the C-terminus of histone 2B) were kindly provided by Dr. Michael M. Goodin (University of Kentucky, Lexington, KY, USA). *N. benthamiana* seedlings were potted in soil and placed in an insect-free growth chamber at 25 °C and 60% relative humidity under a 16 h light/8 h dark photoperiod.

### 2.2. Plasmid Construction

*NbSnRK1 β2* was individually cloned into p2YC, pGADT7, and pGD-GFP vectors, and TbLCYnV *C4* was individually cloned into p2YN, pGBKT7, and pGD-mCherry vectors. TbLCYnV-*C4*-truncated mutants included X1-X10 and TbLCYnV *C4* point mutants containing C4(D22A), C4(S23A), C4(S24A), C4(T25A), C4(W26A), C4(Y27A), C4(P28A), C4(Q29A), C4(P30A), and C4(G31A), which were individually cloned into the pGBKT7 vector. C4 (D22A) was also cloned into a pGD-mCherry vector. A 207 bp fragment of *NbSnRK1 β2* was cloned into a tobacco rattle virus (TRV)-based vector, pTRV-RNA2, to form pTRV2-NbSnRK1 β2. Fragments were PCR-amplified using the KOD One^TM^ PCR master Mix (Toyobo, Shanghai, China). All resulting PCR fragments were fused into vectors using the homologous recombination method by MonClone™ Single Assembly Cloning Mix (Monad, Suzhou, China) and verified by DNA sequencing. All primers used for plasmid construction are listed in Appendix A.

### 2.3. Construction of an Infectious Clone

The construction of the infectious clone of TbLCYnV (pBinPLUS-TbLCYnV-1.5A) and the plasmids pGEM-TbLCYnV-1A and pBinPLUS-TbLCYnV-0.5A were previously described by Xie et al. [9]. pGEM-1A (C4^D22A^) was obtained by introducing the D22A mutation at C4 in pGEM-TbLCYnV-1A using the homologous recombination method (Monad, Suzhou, China). The full-length *Kpn*I-digested fragment of pGEM-1A (C4^D22A^) was inserted into the unique *Kpn*I site of pBinPLUS-TbLCYnV-0.5A, which lacks the C4 region to produce pBinPLUS-TbLCYnV (C4^D22A^)-1.5A, yielding an infectious clone of the mutant virus TbLCYNV (C4^D22A^), as described in previous research [9].

### 2.4. Viral Inoculation and Agroinfiltration

*Agrobacterium* cultures carrying infectious clone TbLCYnV or TbLCYnV (C4^D22A^) were infiltrated into *N. benthamiana* at the 5- to 6-leaf stage for geminivirus agroinoculation. Plants agroinfiltrated with the empty pBinPLUS were used as the mock controls.

For the virus-induced gene silencing (VIGS) assay, *Agrobacterium* cultures harboring pTRV1 or pTRV2-VIGS (pTRV2-GFP or pTRV2-NbSnRK1 β2) were resuspended in an infiltration buffer and mixed at a 1:1 ratio. After a 3 h incubation at room temperature, the mixed *Agrobacterium* cultures were infiltrated into the leaves of *N. benthamiana* plants at the 5- to 6-leaf stage, as described in previous research by Li et al. [22]. Agroinfiltration with pTRV1 and pTRV2-GFP was used as the control. The VIGS efficiency was evaluated using reverse transcription quantitative real-time PCR (qRT-PCR) with the specific primers listed in Appendix A.

### 2.5. Yeast Two-Hybrid (Y2H) Assay

Experimental yeast combinations were co-transformed into a *Saccharomyces cerevisiae* Gold strain according to the Clontech yeast protocol handbook (Takara, Dalian, China). Transformants were grown on synthetic medium SD-Trp/-Leu at 30 °C for 72 h and then transferred to the selective medium SD-Trp/-Leu/-His/-Ade to identify their binding activity.

### 2.6. Bimolecular Fluorescence Complementation (BiFC) Assay

Individual cultures were adjusted to OD_600_ = 1.0 and mixed in equal volumes before leaf infiltration. The suspension was infiltrated into *N. benthamiana* plants at the 7- to 8-leaf stage using 1 mL needleless syringes. The experiments were examined in the epidermal cells of 1–2 cm^2^ leaf explants via confocal microscopy (Leica TCS SP5, Mannheim, Germany) from 36 h to 72 h post infiltration, as described by Li et al. [22].

### 2.7. Co-Immunoprecipitation (Co-IP) Assay

*N. benthamiana* leaves co-infiltrated with experimental combinations at 2 dpi were harvested, ground in 1 mL IP buffer (50 mM Tris-HCl, 150 mM NaCl, 10 mM MgCl_2_, 5 mM DTT, and 0.1% Triton X-100), and centrifuged at 8000× *g* at 4 °C for 15 min. Soluble proteins were cleared by centrifugation and immunoprecipitated with anti-GFP mAb-magnetic beads (MBL, Beijing, China). The Co-IP assay was performed as previously described by Mei et al. [15].

### 2.8. DNA Extraction and Southern Blot Analysis

The total amount of DNA was extracted from infected plant leaves using the cetyltrimethylammonium bromide (CTAB) method by Xin et al. [23]. DNA agarose gels were stained with ethidium bromide to provide a loading control. After denaturation and neutralization, all DNA samples were transferred to Hybond N+ nylon membranes (GE Healthcare, Pittsburgh, PA, USA) by capillary transfer. After that, the viral genomic DNA samples were hybridized with a coat protein digoxin-labeled probe using a DIG High Prime DNA Labeling and Detection Starter Kit II (Roche Diagnostics, Mannheim, Germany), as described in previous research by Mei et al. [15].

### 2.9. RNA Extraction, Quantitative RT-PCR (qRT-PCR), and qPCR Analysis

The total amount of RNA was extracted from *N. benthamiana* leaf tissues using Trizol reagent (Invitrogen, Carlsbad, CA, USA) according to the manufacturer’s instructions. cDNA was synthesized using ReverTra Ace qPCR RT Master Mix with gDNA Remover (Toyobo, Osaka, Japan). qPCR was performed using ChamQ SYBR Color qPCR Master Mix (Vazyme, Nanjing, China), and expression data were normalized with the expression level of *NbActin*, as described in previous research by Li et al. [22]. The viral DNA levels were determined by qPCR, and all the specific primers are listed in Appendix A.

### 2.10. Protein Extraction and Western Blotting

Plant total protein extraction and Western blotting were performed as previously described by Li et al. [22]. Immunoblotting was performed with primary antibodies: Mouse anti-GFP-Tag mAb antibody (ABclonal, Wuhan, China), Mouse anti-mCherry-Tag mAb antibody (ABclonal, Wuhan, China), and then HRP* Goat anti-Mouse IgG antibody (ImmunoWay, Suzhou, China). Blotted membranes were washed thoroughly and visualized using chemiluminescence according to the manufacturer’s protocol (4A biotech, Suzhou, China).

### 2.11. Chemical Treatments

The inhibitors of protein degradation pathways infiltrated into the plant leaves. MG132 (a specific 26S proteasome inhibitor) (MCE, Shanghai, China), 3-MA (an inhibitor of autophagy via its inhibitory effect on class III PI3K) (MCE), and E64d (a cysteine protease inhibitor associated with autophagy) (MCE) (all dissolved in DMSO) were diluted in ddH_2_O to 100 μM, 5 mM, and 100 μM, respectively [24,25,26]. An equal volume of DMSO was diluted in ddH_2_O as the control. To inhibit protein synthesis, CHX (protein synthesis inhibitor) (MCE) dissolved in ethanol was added to 100 µM. After infiltration, leaves were moisturized for 4 h and then harvested. Chemical treatment assays were performed as described by Li et al. [22].

## 3. Results

### 3.1. A Subunit β2 of SnRK1 Is Screened to Interact with TbLCYnV C4

We performed a Y2H assay to identify the cellular factors targeted by the TbLCYnV C4 protein. Briefly, we fused TbLCYnV C4 to the GAL4 DNA-binding domain (BD) as bait and identified an *N. benthamiana* cDNA library for the GAL4 activation domain (AD). Double transformants able to grow in SD-Trp/-Leu/-His were screened, and we recovered one cDNA clone that matched a SnRK1 regulatory subunit β2 (Sequence ID: Niben101Scf00797g17008.1), which we named NbSnRK1 β2. Data analysis showed that the full length of NbSnRK1 β2 contained 846 nucleotides. The phylogenetic tree also showed that NbSnRK1 β2 clustered with SnRK1 β2 in *N. tomentosiformis*, *N. tabacum*, *N. attenuata*, and *N. sylvestris* and formed different branches with other SnRK1 β2 in *Solanum stenotomum*, *S. tuberosum*, *S. verrucosum*, *S. lycopersicum*, *S. pennellii*, *Capsicum annuum*, *Lycopersicon esculentum*, *Coffea arabica*, *C. eugenioides*, *Hibiscus syriacus*, *Salvia hispanica*, *Ipomoea nil,* and *I. triloba* (Figure 1A and Appendix A). Using *interpro* (https://www.ebi.ac.uk/interpro, accessed on 9 June 2023), we predicted that the NbSnRK1 β2 protein had 282 amino acid (aa) and contained two conserved domains: carbohydrate-binding module domain (CBM, 94-174 aa) and association with the SnRK1 kinase complex domain (ASC, 189-279 aa) (Figure 1B). The CBM domain can bind to starch or sucrose substrates, and the ASC domain is responsible for forming the SnRK1 complex [27].

### 3.2. TbLCYnV C4 Interacts with NbSnRK1 β2 In Vitro and In Vivo

To validate the interaction between TbLCYnV C4 and NbSnRK1 β2, we first performed the Y2H assay. The full-length open reading frame (ORF) encoding NbSnRK1 β2 was fused to AD. T7 was co-expressed with P53 and lam as the positive and negative control, respectively. We found that the strain carried combinations of AD-NbSnRK1 β2/BD-C4 could grow on the SD-Trp/-Leu/-His/-Ade medium like the positive control, while the strains that carried AD-NbSnRK1 β2/BD-empty or AD-empty/BD-C4 could not (Figure 2A).

We also conducted a BiFC assay in vivo. TbLCYnV C4 and NbSnRK1 β2 were fused to split the N- or C-terminal parts of the yellow fluorescent protein (YFP), respectively, and expressed in *H2B-RFP* transgenic *N. benthamiana* plants. Confocal microscopy detected the presence of YFP fluorescence in both cytoplasm and nuclear, indicating that NbSnRK1 β2 and TbLCYnV C4 could interact in these two subcellular compartments (Figure 2B). The TbLCYnV C4–NbSnRK1 β2 interaction was also confirmed by Co-IP assay. C4-mCherry was co-expressed with NbSnRK1 β2-GFP or GFP in *N. benthamiana* leaves and then subjected to immunoprecipitation using anti-GFP mAb-magnetic beads, and Co-IP products were detected with anti-GFP and anti-mCherry antibodies. Again, a specific interaction between TbLCYnV C4 and NbSnRK1 β2 but not TbLCYnV C4 and GFP was observed in the Co-IP assay (Figure 2C). Taken together, these results suggest that the TbLCYnV C4 protein directly interacts with the NbSnRK1 β2 protein both in vitro and in vivo.

### 3.3. Silencing of NbSnRK1 β2 Enhances the Susceptibility of N. benthamiana to TbLCYnV Infection

To better understand the role of NbSnRK1 β2 in the plant and the potential effect of NbSnRK1 β2 on the function of viral infection, we silenced *NbSnRK1 β2* in *N. benthamiana* by using the VIGS assay with a TRV-based vector. Part of the coding sequence (511–717 nt) of NbSnRK1 β2 was cloned into pTRV2-RNA to form pTRV2-NbSnRK1 β2. *Agrobacterium tumefaciens* carrying recombinant pTRV2-VIGS (pTRV2-NbSnRK1 β2 or pTRV2-GFP) was mixed with that carrying pTRV1 before being inoculated into *N. benthamiana*. *NbSnRK1 β2* silencing in *N. benthamiana* plants (TRV-NbSnRK1 β2) did not result in any distinct developmental defects in systemic leaves compared to GFP-silenced plants (TRV-GFP) as negative controls. qRT-PCR analysis revealed that the transcripts of *NbSnRK1 β2* were significantly reduced (over 70%) in TRV-NbSnRK1 β2-silenced *N. benthamiana* plants compared to those in *N. benthamiana* plants inoculated with the control (TRV-GFP) (Figure 3B). Then, TbLCYnV was inoculated in TRV-NbSnRK1 β2-silenced and GFP-silenced *N. benthamiana* plants, and the empty pBinPLUS vector was also inoculated in these two kinds of plants as Mock. Infection induced by TbLCYnV in TRV-NbSnRK1 β2-inoculated plants showed more severe symptoms like leaf up-curling and dwarfing than those of TRV-GFP-inoculated plants at 20 or 40 days post incubation (dpi) (Figure 3A). As expected, qPCR and Southern blot analyses of viral genomic DNA levels indicated that higher levels of TbLCYnV genomic DNA were found in TRV-NbSnRK1 β2-inoculated plants than in TRV-GFP-inoculated plants (Figure 3C,D). These results indicate that *NbSnRK1 β2* silencing enhanced the susceptibility of *N. benthamiana* to TbLCYnV infection, suggesting that NbSnRK1 β2 might act as a disease resistance factor and play an important role in disease symptom development and viral accumulation through NbSnRK1 β2–TbLCYnV C4 interactions.

### 3.4. Asp22 of TbLCYnV C4 Is Vital for TbLCYnV C4–NbSnRK1 β2 Interaction and Viral Infection

To explore which region in TbLCYnV C4 is responsible for the interaction with NbSnRK1 β2, ten C4-truncated mutants were constructed based on the full-length 97 aa of C4, and each mutant was deleted 10 aa in the C4 X1–X9, while 5 aa (positions 92–96) were deleted in the C4-X10 mutant (Figure 4A). Then, these ten C4-truncated mutants were fused to the BD, and they were co-transformed in yeast with AD-NbSnRK1 β2, respectively. The Y2H assays showed that all mutants except C4-X3 (22–31 aa) interacted with NbSnRK1 β2 (Figure 4B). These results demonstrate that the 22–31 amino acid position of C4 is indispensable for its interaction with NbSnRK1 β2.

Next, we carried out site-directed mutagenesis to pinpoint the key residue involved in the interaction. Every single amino acid at positions 22–31 was replaced with alanine (Ala, A.); then, we conducted a Y2H assay (Figure 4C,D). Notably, all point mutants except C4 (D22A) interacted with NbSnRK1 β2. These results indicate that aspartic acid (Asp, D) at position 22 aa of C4 is a key site for the TbLCYnV C4–NbSnRK1 β2 interaction.

To clarify the effect of C4 (D22A) on the pathogenicity of the virus, the infectious clone of the mutant virus was constructed. The point mutant C4 (D22A) was introduced into the pBinPLUS vector to form pBinPLUS-TbLCYnV (C4^D22A^)-1.5A, yielding an infectious clone of the mutant virus TbLCYNV (C4^D22A^). *A. tumefaciens* carrying infectious clone TbLCYnV (C4^D22A^) or wild TbLCYnV were inoculated into *N. benthamiana* leaves. Notably, *N. benthamiana* plants infected by the wild TbLCYnV showed more severe symptoms than the TbLCYnV (C4^D22A^)-infected plants. At 10 dpi, the upper leaves of TbLCYnV (C4^D22A^)-infected plants were slightly curled when those of TbLCYnV-infected plants were completely up-curled. The TbLCYnV-infected plants produced severe up-curling, vein thickening, and dwarfing symptoms at 30 dpi, while only 2–3 upper leaves in the TbLCYnV (C4^D22A^)-infected plants showed curling symptoms (Figure 4E). As expected, the qPCR analysis of the viral genomic DNA levels indicated that lower levels of genomic DNA were found in the mutant TbLCYnV (C4^D22A^)-infected plants compared to the wild TbLCYnV-infected *N. benthamiana* plants, and no TbLCYnV signal could be detected in control plants (Figure 4F). Southern blot analysis showed that TbLCYnV accumulation noticeably decreased in TbLCYnV (C4^D22A^)-infected plants compared to the TbLCYnV-infected plants (Figure 4G). These results point out that TbLCYnV (C4^D22A^)-infected plants show milder symptoms, and the viral accumulation is significantly reduced compared to that of the wild virus in *N. benthamiana*, indicating that the pathogenicity of the mutant virus TbLCYnV (C4^D22A^) decreased.

### 3.5. TbLCYnV C4 Promotes the Degradation of NbSnRK1 β2 Via the Autophagy Pathway

To detect whether TbLCYnV C4 affects the NbSnRK1 β2 expression, we co-expressed NbSnRK1 β2-GFP with C4-mCherry or mCherry in *N. benthamiana* leaves and found a lower level of NbSnRK1 β2 protein expression in the C4-mCherry-inoculated *N. benthamiana* leaves (0.75) compared with that of the mCherry-inoculated *N. benthamiana* leaves (1.00) according to Western blot (Figure 5A). Semi-qRT-PCR assay and qRT-PCR assay showed that there was no significant difference in the transcription level of *NbSnRK1 β2* in *N. benthamiana* between the two treatments (Figure 5A and Appendix A). This demonstrates that TbLCYnV C4 may reduce NbSnRK1 β2 protein accumulation. Next, we verified the same results using three other independent experiments and performed statistical analysis using GraphPad Prism 8 (Figure 5B). These results suggest that the TbLCYnV C4 can cause NbSnRK1 β2 degradation.

To investigate the potential pathways involved in NbSnRK1 β2 degradation, NbSnRK1 β2-GFP-inoculated *N. benthamiana* leaves were treated with 100 μM CHX and 100 μM E64d solution (autophagy inhibitors), 5 mM 3-MA solution (autophagy inhibitors), 100 μM MG132 solution (proteasome inhibitor), or DMSO solution as the control treatment, respectively. The Western blot results showed that higher levels of NbSnRK1 β2 protein expression were detected in the E64d-treated (2.01) and 3-MA-treated (1.51) leaves than in the DMSO-treated (1.00) leaves. In contrast, the signals of NbSnRK1 β2 protein expression were almost the same in the MG132-treated (1.02) and the DMSO-treated (1.00) leaves (Figure 5C). These results suggest that NbSnRK1 β2 protein is degraded through the autophagy pathway.

Then, we co-expressed NbSnRK1 β2-GFP with C4-mCherry or mCherry in *N. benthamiana* leaves, and the leaves were treated with the 100 μM E64d solution or DMSO solution as the control treatment. The Western blot results showed that the accumulation of NbSnRK1 β2 protein in the NbSnRK1 β2- and C4-mCherry co-infiltrated plants (0.71, 0.34) was lower than those in the NbSnRK1 β2- and mCherry co-infiltrated plants (1.49, 1.00). With E64d treatment, the accumulation of NbSnRK1 β2 protein (0.71, 1.49) was obviously higher than in DMSO-treated leaves (0.34, 1.00), while the C4 expression level was almost same in E64d or DMSO treatments (Figure 5D). These results indicate that NbSnRK1 β2 was degraded through the autophagy pathway, and TbLCYnV C4 affected its degradation via C4–NbSnRK1 β2 interaction.

To further examine whether the C4 mutant can cause NbSnRK1 β2 degradation, we conducted semi-in-vivo degradation assays to agroinfiltrate NbSnRK1 β2-GFP with C4 (D22A)-mCherry or C4-mCherry in *N. benthamiana* leaves for 48 h and then treated the samples with 100 μM E64d. The Western blot results showed that the level of NbSnRK1 β2 protein expression in C4 (D22A)-mCherry-inoculated *N. benthamiana* leaves was higher (1.79) than in C4-mCherry-inoculated *N. benthamiana* leaves (1.00), while the expression of C4 was almost same in the C4 (D22A)-mCherry and C4-mCherry inoculated plants (Figure 5E). These results indicate that C4 can cause more NbSnRK1 β2 degradation than the mutant C4 (D22A). These results further confirm that C4 promotes NbSnRK1 β2 degradation via the autophagy pathway based on the TbLCYnV C4–NbSnRK1 β2 interaction.

## 4. Discussion

Geminiviruses are a group of plant viruses that cause severe agricultural losses worldwide [28]. In geminivirus–plant arms races, plants have developed effective strategies targeting viral infection steps. In response, geminiviruses encode multifunctional proteins like C4 to hijack specific host factors to antagonize host defenses [29]. C4 can target many host proteins, such as Kip-related proteins (KRP), barely any meristem 1(BAM1), and BRI1 kinase inhibitor1 (BKI1), to facilitate viral infection [30,31,32]. In this study, we discovered that TbLCYnV C4 could interact with a new host protein, namely NbSnRK1 β2, a regulatory subunit of SnRK1. The Y2H, BiFC, and Co-IP assays showed that TbLCYnV C4 interacted with NbSnRK1 β2 *in vitro* and *in vivo* (Figure 2), and the key site of interaction was Asp at position 22 in C4 (Figure 4A–D). Then, the infectious clone of the mutant virus TbLCYnV (C4^D22A^) was constructed and produced milder symptoms. The viral accumulation significantly decreased than that of the wild virus in *N. benthamiana* (Figure 4E–G), indicating that the pathogenicity of the mutant virus TbLCYnV (C4^D22A^) was attenuated. Meanwhile, the VIGS assay showed that NbSnRK1 β2 might act as a plant defense factor to interfere with the TbLCYNV infection (Figure 3).

It has been reported that SnRK1 plays a defensive role in plant antiviral defenses [19]. In tobacco, the overexpression or antisense expression of SnRK1 α enhances or weakens plant resistance to geminivirus infection [33]. Beet curly top virus (BCTV) C2 may indirectly suppress the antiviral defenses of SnRK1 α during geminivirus infection by interacting with adenosine kinase (ADK) and inactivating ADK [34]. SnRK1 protein attenuates geminivirus infection by interacting with the βC1 protein of tomato yellow leaf curl China betasatellite (TYLCCNB) and phosphorylating the βC1 protein [35]. AL2/C2 proteins from tomato mottle virus (ToMoV), tomato golden mosaic virus (TGMV), tomato yellow leaf curl virus (TYLCV), and BCTV are all phosphorylated by SnRK1 α, resulting in a delay in viral infection [36]. Our results indicated that TbLCYnV C4 might disturb the resistance function of NbSnRK1 β2 in some way and affect the infection of TbLCYnV.

It has been reported that viruses can utilize some strategies to weaken the plant defenses by degrading host proteins. The viral genome-linked protein (VPg) of turnip mosaic virus (TuMV) binds to the suppressor of gene silencing 3 (SGS3) and RNA-dependent RNA polymerase 6 (RDR6) to induce their degradation [37]. In this study, treatment with autophagy inhibitors E64d or 3-MA inhibited NbSnRK1 β2 degradation, which indicates that NbSnRK1 β2 is degraded through the autophagy pathway (Figure 5C). The co-infiltration experiment of NbSnRK1 β2 and C4 showed that TbLCYnV C4 promoted NbSnRK1 β2 degradation (Figure 5A,B,D), while NbSnRK1 β2 degradation was reduced by the C4 (D22A) mutant (Figure 5E), suggesting that the interaction of TbLCYnV C4–NbSnRK1 β2 mediated NbSnRK1 β2 degradation. During energy starvation, SnRK1 α is activated to induce autophagy by phosphorylating autophagy-related protein 1 (ATG1), ATG6, and FYVE domain proteins required for endosomal sorting 1 (FREE1) or inhibiting target-of-rapamycin (TOR) signaling. The activation of autophagy, in turn, accelerates the degradation of FCS-like zinc finger (FLZ) proteins that can inhibit SnRK1 signaling, which further activates SnRK1 and helps plants to better adapt to stress [38]. The C4 protein encoded by cotton leaf curl Multan virus (CLCuMuV) inhibits autophagy by binding to the autophagy negative regulator (eIF4A) to enhance the eIF4A-autophagy-related protein 5 (ATG5) interaction [39]. TbLCYnV C4 might utilize ATGs to regulate the autophagy pathway, while the specific mechanism by which C4 promotes NbSnRK1 β2 degradation remains to be further studied.

Understanding the role of SnRK1 in host defense is important for disease resistance breeding in crops. Studies about SnRK1 mainly focus on SnRK1 α, while few studies have investigated the subunit β. In this study, we found that a subunit β of SnRK1, namely NbSnRK1 β2, is a plant defense factor able to withstand the infection of geminivirus TbLCYnV. During viral infection, TbLCYnV C4 interacted with NbSnRK1 β2 and promoted the degradation of the latter by inhibiting a plant defense system to facilitate viral infection. It is well known that the subcellular localization of a viral protein is highly relevant to its functions during viral infection. After artificially activating pattern-triggered immunity (PTI), the C4 protein from TYLCV can shift from the plasma membrane to the chloroplasts, interfering with the chloroplast-dependent anti-viral salicylic acid (SA) biosynthesis to promote virulence [39]. NbSnRK1 β2 contains two conserved domains: the ASC domain, which is responsible for the formation of the SnRK1 complex, and the CBM domain, which can bind to starch or sucrose substrates. We found it interesting that the key domain in NbSnRK1 β2 interacts with TbLCYnV C4, which might mediate C4 to its precise localization to perform its other functions against plant defense or affect the assembly of the NbSnRK1 complex to reduce the host antiviral defense. Using the Y2H assay, we found that TbLCYnV C4 only interacted with NbSnRK1 β2 but not NbSnRK1 α (Appendix A), suggesting that TbLCYnV C4 is a unique protein that, unlike the C2 or βC1 which interacts with the NbSnRK1 α and can be phosphorylated by NbSnRK1 α. The relationship of these two subunits, i.e., NbSnRK1 β2 and NbSnRK1 α, and whether NbSnRK1 β2 degradation regulated by TbLCYnV C4 would affect the global function of SnRK1 in *N. benthimiana* need further study.

## 5. Conclusions

In summary, the TbLCYnV C4 protein facilitates viral infection by interacting with NbSnRK1 β2 and promoting its degradation to disturb plant defenses, which may provide new insights into plant–geminivirus interactions.

## Figures and Tables

**Figure 1 viruses-16-00234-f001:**
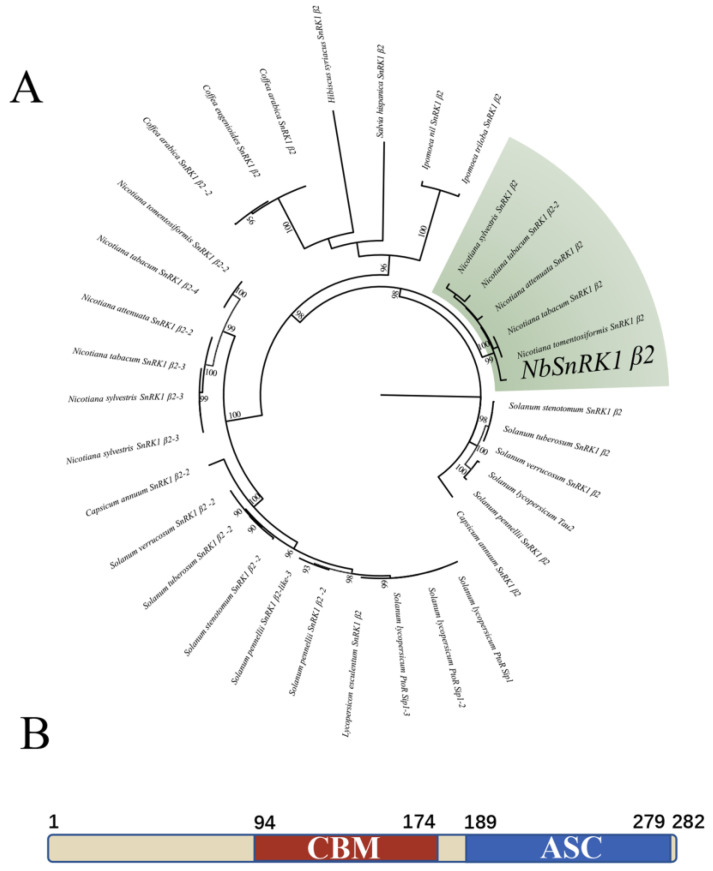
Phylogenetic tree (**A**) and schematic representation (**B**) of NbSnRK1 β2. (**A**) Phylogenetic tree based on the full-length nucleotide sequences of NbSnRK1 β2 shown in Appendix A. The nucleotide sequence alignment was performed in ClustalX2, and the phylogenetic tree was created with 1000 replicates by neighbor-joining method in the MEGA software 11.0.9. Bootstrap scores over 90% were placed at major nodes. (**B**) Schematic diagram showing the NbSnRK1 β2 protein structure. CBM domain, carbohydrate-binding module; ASC domain, association with SNF1 kinase complex. Numbers represented the amino acid positions of NbSnRK1 β2 protein.

**Figure 2 viruses-16-00234-f002:**
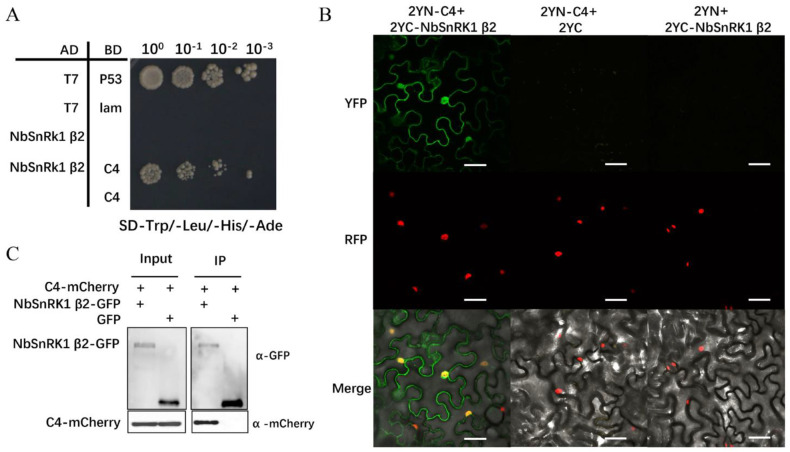
Demonstrated interaction of TbLCYnV C4 with NbSnRK1 β2 in vitro and in vivo. (**A**) Interaction between TbLCYnV C4 and NbSnRK1 β2 by Y2H assay. Yeast strains of Gold co–transformed with the indicated plasmids were subjected to 10–fold serial dilutions and grown on an SD-Trp/-Leu/-His/-Ade medium. The strain–carried combinations of AD-T7/BD-P53 were used as a positive control, and those with AD-T7/BD-lam or with empty vectors BD and AD were used as negative controls. (**B**) BiFC assay showing the interaction between TbLCYnV C4 with NbSnRK1 β2 in vivo. The *H2B-RFP* transgenic *N. benthamiana* leaves co-expressing 2YN-C4 and 2YC-NbSnRK1 β2, 2YN-C4 and 2YC, or 2YN and 2YC-NbSnRK1 β2 were examined and imaged under confocal microscope. Expression of H2B-RFP was used a nuclear marker. Columns from top to bottom represent YFP, red fluorescence (RFP), and YFP/RFP merged field (merge). Scale bars represent 50 μm. (**C**) Interaction between C4 and NbSnRK1 β2 by Co-IP assay in vivo. *N. benthamiana* leaves were co-infiltrated TbLCYnV C4 with NbSnRK1 β2-GFP or GFP. Total proteins were extracted from the infiltrated leaf samples at 2 dpi. The samples were carried out with anti-GFP mAb-magnetic beads. The input and the co–immunoprecipitated proteins were analyzed by Western blot using anti-GFP or anti-mCherry antibody.

**Figure 3 viruses-16-00234-f003:**
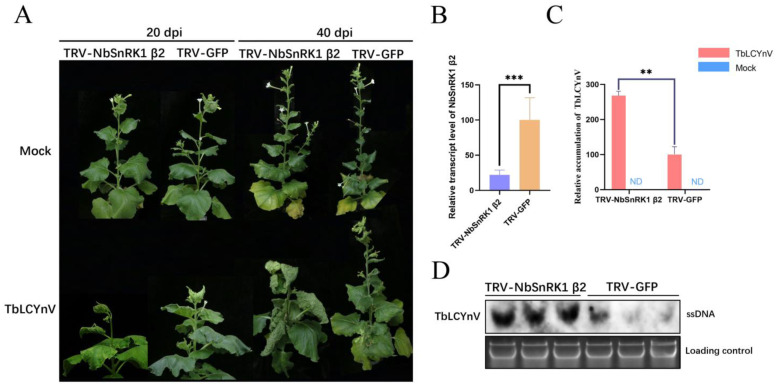
NbSnRK1 β2-silenced *N. benthamiana* plants were susceptible to TbLCYnV infection. (**A**) Symptoms of the *N. benthamiana* plants infected by TbLCYnV in TRV-NbSnRK1 β2- or TRV-GFP-inoculated plants at 20/40 dpi. (**B**) qRT-PCR analysis showing the expression efficiency of NbSnRK1 β2 in TRV-NbSnRK1 β2- or TRV-GFP-inoculated plants. The values represent relative NbSnRK1 β2 mRNA levels in TRV-NbSnRK1 β2-inoculated seedlings against the control TRV-GFP-inoculated seedlings, and the data of the TRV-GFP were set as 100. *NbActin* is used as an internal reference. The data are shown as means and standard deviation (SD) of six biological replicates. *** indicates a highly significant difference between the two treatments at the *p* < 0.001 by Student’s *t*-test. (**C**) Viral accumulation in *N. benthamiana* plants shown in panel A was determined by qPCR at 40 dpi. The values represent viral DNA accumulation in TbLCYnV-infected NbSnRK1β2-or GFP-silenced plants, and the data of the TRV-GFP were set as 100. ND, not detectable in control plants. ** indicates a significant difference between the two treatments at *p* < 0.01 by Student’s *t*-test. The data are shown as means and SD of three biological replicates. (**D**) Southern blot analysis of the TbLCYnV accumulation in TRV-NbSnRK1 β2- or TRV-GFP-inoculated *N. benthamiana* plants. Total genomic DNA (approximately 15 ug for each lane) from a mixture of three seedlings was extracted at 40 dpi. The blot was probed with the coat protein of TbLCYnV. An ethidium-bromide-stained gel shown below the blots provided a DNA loading control. The position of the single-stranded DNA (ssDNA) form is indicated.

**Figure 4 viruses-16-00234-f004:**
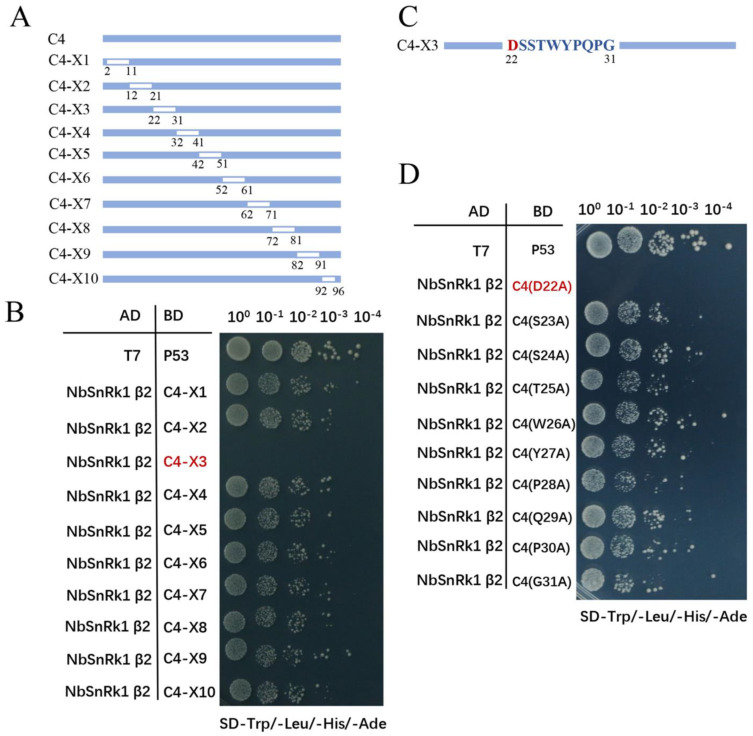
Asp22 of TbLCYnV C4 is vital for C4–NbSnRK1 β2 interaction and viral infection. (**A**) Schematic representation of the ten truncated mutants of TbLCYnV C4 (C4-X1 to X10). Each mutant was deleted ten amino acids in the X1-X9, while five amino acids (92–96) were deleted in the X10 mutant. (**B**) Interaction of the ten truncated mutants of TbLCYnV C4 with NbSnRK1 β2 tested by Y2H assay. Yeast strains of Gold co-transformed with the indicated plasmids were subjected to 10-fold serial dilutions and grown on an SD-Trp/-Leu/-His/-Ade medium. AD-T7/BD-P53 was used as a positive control, and that with AD-T7/BD-lam was used as a negative control. (**C**) Schematic diagram of the ten-point mutants of TbLCYnV C4. Each single aa at positions 22–31 was replaced with Ala. (**D**) Interaction of the ten-point mutants of TbLCYnV C4 shown in panel C with NbSnRK1 β2 tested by Y2H assay. (**E**) Symptoms of *N. benthamiana* plants infected with TbLCYnV or the mutant TbLCYnV (C4^D22A^). The plants were inoculated with *Agrobacterium tumefaciens* carrying wild TbLCYnV or infectious clone TbLCYnV (C4^D22A^) or with an empty pBinPLUS vector as mock. Photographs were taken at 10/30 dpi. (**F**) TbLCYnV DNA accumulation in *N. benthamiana* plants shown in panel E by qPCR at 30 dpi. The values represent viral DNA accumulation in TbLCYnV (C4^D22A^) -infected or TbLCYnV-infected plants, and the data of TbLCYnV-infected plants were set as 100. ND, not detectable in control plants. ** indicates a significant difference between the treatments at *p* < 0.01 by Student’s *t*-test. The data are shown as means and SD of three biological replicates. (**G**) TbLCYnV DNA accumulation in *N. benthamiana* plants shown in panel E by Southern blot at 30 dpi. Total genomic DNA (approximately 10 ug for each lane) from a mixture of three seedlings was used in Southern blot. The blot was probed with the coat protein of TbLCYnV. An ethidium-bromide-stained gel shown below the blots provided a DNA loading control. The position of single-stranded DNA (ssDNA) form is indicated.

**Figure 5 viruses-16-00234-f005:**
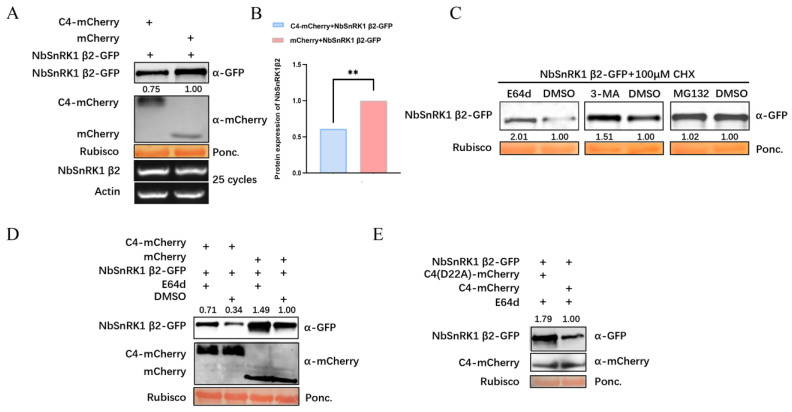
TbLCYnV C4 promotes the degradation of NbSnRK1 β2 via the autophagy pathway. (**A**) Western blot analysis and semi-qRT-PCR assay of NbSnRK1 β2 protein and mRNA levels. *N. bent hamiana* leaves co-expressed NbSnRK1 β2-GFP with C4-mCherry or mCherry, respectively. Samples were analyzed by immunoblot using anti-GFP or anti-mCherry antibody. The amount of NbSnRK1 β2-GFP expression was calculated by ImageJ 1.8.0, and the data of NbSnRK1 β2-GFP co-infiltrated with mCherry were set as 1.00. Ponceau Red Stained Rubisco was used as a loading control. Semi-qRT-PCR assay was used to detect the transcription levels of *NbSnRK1 β2* in plants. The expression of *Actin* gene was used as an internal control. (**B**) The expression of NbSnRK1 β2-GFP was calculated by GraphPad Prism 8. The data are shown as means and SD of three biological replicates. ** indicates a significant difference between the two treatments at the *p* < 0.01 by Student’s *t*-test. (**C**) Protease inhibitor treatment of NbSnRK1 β2 by Western blot. The *N. benthamiana* leaves transiently expressing NbSnRK1 β2-GFP were treated with 100 μM CHX together with 100 μM E64d, 5 mM 3-MA, 100 μM MG132, or DMSO, respectively. Samples were analyzed by immunoblot using anti-GFP antibody. The relative accumulation level of NbSnRK1 β2 in DMSO treatments were set as 1.00. Ponceau Red Stained Rubisco was used as a loading control. (**D**) The expression levels of NbSnRK1 β2-GFP by Western blot. The *N. benthamiana* leaves transiently co-expressing C4-mCherry or mCherry with NbSnRK1 β2-GFP were treated with 100 μM E64d or DMSO, respectively. The amount of NbSnRK1 β2-GFP expression was calculated using ImageJ 1.8.0. The relative accumulation level of NbSnRK1 β2 in the co-infiltrated with NbSnRK1 β2 and mCherry plants with DMSO treatment was set at 1.00. Ponceau Red Stained Rubisco was used as a loading control. (**E**) Semi-in-vivo protein stability assay of NbSnRK1 β2 by Western blot. NbSnRK1 β2 was transiently co-expressed with C4 (D22A)-mCherry or C4-mCherry in *N. benthamiana* plants, and E64d treatment was performed 48 h later. The data of NbSnRK1 β2-GFP co-infiltrated with C4-mCherry were set as 1.00. Ponceau Red Stained Rubisco was used as a loading control.

## Data Availability

Data are contained within the article and Appendix A.

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
