# Peer review of "The C4 Protein of TbLCYnV Promotes SnRK1 β2 Degradation Via the Autophagy Pathway to Enhance Viral Infection in N. benthamiana"

_viruses, 2024, doi:10.3390/v16020234_

Round 1

Reviewer 1 Report

Comments and Suggestions for Authors

The manuscript “C4 protein of TbLCYnV promotes SnRK1 β2 degradation via the autophagy pathway to enhance viral infection in N. benthamiana”, describes the interaction between the TbLCYnV C4 protein and the N. benthamiana β2 subunit of SnRK1 kinase. The authors show that silencing of SnRK1 β2 increases N. benthamiana susceptibility more to TbLCYnV wt than to a viral mutant unable to interact with SnRK1 β2. Furthermore, they show that Nb SnRK1 β2 accumulates in lower amounts in the presence of C4 when both proteins are co-expressed in N. benthamiana leaves and in higher amounts in the presence of the autophagy inhibitors E64d and 3-MA. Finally, the authors compare Nb SnRK1 β2 accumulation in the presence of C4 wt and D22A mutant claiming that in the presence of the latter Nb SnRK1 β2 is less abundant.

I find the results illustrated in Figures 1-4 robust and generally well-explained and I agree with the conclusions the authors draw on this part of the manuscript. However, short explanations of critical materials and techniques utilized and of the logic of the experiment, would facilitate comprehension, e.g line 220, explain what H2B-RFP transgenic N. benthamiana plants are and why they are used. Please, provide information on the number of biological replicates used in the RT-qPCR experiment and the house keeping gene(s) utilised.

On the other hand, I have a few doubts about some of the results illustrated in Fig 5, and particularly about their discussion.

Figure 5A. Line 352; since transcript quantification is not determined, the performed experiment does not allow to prove whether C4 affects SnRK1 β2 expression upon co-expression. It rather shows that C4 binds to SnRK1 β2 thus reducing its accumulation in N. benthamiana leaves.

Figure 5D. The authors claim (lines 378-379) that “NbSnRK1 β2 is degraded through the autophagy pathway within the interaction with TbLCYnV C4”. While NbSnRK1 β2 accumulation is reduced in the presence of C4-mCherry (Fig 5A), treatment with E64d, inhibiting the autophagy degradation pathway, favours NbSnRK1 β2-GFP accumulation. However, to claim a possible role of C4-mCherry in such a process, the authors should investigate the accumulation of NbSnRK1 β2-GFP in the same conditions (with and without E64d) in the absence of C4-mCherry. Indeed, C4-mCherry seems to be less abundant in the presence of NbSnRK1 β2-GFP (compare the third lane with the first and second lanes) suggesting that it is partially sequestered in the binding to NbSnRK1 β2-GFP. However, this does not prove that this interaction is impacting NbSnRK1 β2-GFP degradation.

Also, an additional control test without C4-mCherry is necessary to rule out that the tag of mCherry does not affect the expression of C4.

Figure 5E. The difference in NbSnRK1 β2-GFP accumulation in the presence of the C4(D22A) mutant compared to the C4wt is hard to detect. It is so faint that it might be due to a slight difference in sample loading. Also, the effect of the D22A mutation of C4-mCherry compared to the wt C4-mCherry on NbSnRK1 β2 accumulation should be tested in the presence of E64d.

In conclusion, I think that taken together these results do not provide sufficiently strong evidence to claim an active role of C4 in the NbSnRK1 β2 degradation, and that the C4 -NbSnRK1 β interaction directly reduces plant defence.

Finally, in my opinion, the discussion contains too much information already included in the introduction and many repetitions. On the other hand, it is missing discussion on the possible mechanisms of C4 action particularly in the context of the SnRK1 heterotrimeric complex.

It would benefit from extensive editing.

Reviewer 2 Report

Comments and Suggestions for Authors

The authors need to re-edit the manuscript comprehensively to make the English more readable. I cannot review the content without getting annoyed constantly. Listed below are problems spotted in Abstract only.

Ln15, “their infections”

Ln18, change “Nowadays, studies…” to “Previous studies”

Ln19, “diseases” should be “microbial pathogens” or “biotic stresses”

Ln19-20, delete “while…” you don’t need to be comprehensive in Abstract.

Ln20, change “C4 of” to “the C4 protein encoded by”

Ln21, add “Nicotiana benthamiana” before “SnRK1”.

Ln23, end the sentence after “TbLCYnV infection. Start a new sentence with “Plants infected with…”

Ln26, delete “Here”.

Ln27, change “a novel insight” to a model”.

Ln27-29, please re-write. This is essentially a repetition of the previous sentences.

Comments on the Quality of English Language

Reviewer 3 Report

Comments and Suggestions for Authors

In this MS the authors provide evidence on the interaction of C4 protein of TbLCYnV with the NbSnRK1 β2 subunit in vivo and in planta. This interaction promotes the degradation of the NbSnRK1 β2 protein via the autophagy-mediated mechanism and results in the regulation of downstream plant defenses. Silencing of NbSnRK1 β2 in N. benthamiana plants results in enhanced viral accumulation and symptom development. Furthermore, they showed that the Asp22 residue in C4 protein is critical for this interaction. Overall, the MS is well written, the experiments are well designed and performed, and therefore it merits publication in Viruses. Some minor comments can be found in the attached file.

Comments on the Quality of English Language

Minor editing on English language is needed.

Round 2

Reviewer 1 Report

Comments and Suggestions for Authors

I am pleased with the new experiments performed by the authors and I think that taken together the presented results support the hypothesis that the TbLCYnV C4- NbSnRK1 β2 interaction can interfere with the NbSnRK1 β2 degradation pathway. In the manuscript, the authors refer to this as ( “suggesting that the interaction of TbLCYnV C4–-NbSnRK1 β2 mediated 548

the degradation of NbSnRK1 β2 degradation”.e(.g. line 548-549). As proved in Figure 5C, NbSnRK1 β2 can be degraded in the absence of TbLCYnV C4, therefore, the used term “mediates” overstates this phenomenon and should be replaced in the whole text.

On the other hand, the RT-PCR result that the authors have included in Figure 5A cannot prove the transcriptional level of SnRK1 β2 in the two experimental conditions. The authors should be aware that PCR is not a quantitative assay, this result only proves that the transcripts are present in both samples. Quantitative RT-PCR or Northern-blot analyses on the same samples used for the Western-blot can investigate the level of expression of the coding gene.

English usage in this manuscript must be substantially improved.

Comments on the Quality of English Language

English usage in this manuscript must be substantially improved.
